# Differences in Blood-Derived *Francisella tularensis* Type B Strains from Clinical Cases of Tularemia

**DOI:** 10.3390/microorganisms8101515

**Published:** 2020-10-01

**Authors:** Marilynn A. Larson, Baha Abdalhamid, Bhanwar Lal Puniya, Tomáš Helikar, David W. Kelley, Peter C. Iwen

**Affiliations:** 1Department of Pathology and Microbiology, University of Nebraska Medical Center, Omaha, NE 68198, USA; babdalhamid@unmc.edu (B.A.); david.kelley@unmc.edu (D.W.K.); piwen@unmc.edu (P.C.I.); 2Department of Biochemistry, University of Nebraska-Lincoln, Lincoln, NE 68588, USA; bpuniya2@unl.edu (B.L.P.); thelikar2@unl.edu (T.H.); 3Nebraska Public Health Laboratory, Omaha, NE 68198, USA

**Keywords:** *Francisella tularensis* subsp. *holarctica* (type B), human tularemia cases, human blood-derived *F. tularensis*, single nucleotide polymorphisms

## Abstract

*Francisella tularensis* can cause the zoonotic disease tularemia and is partitioned into subspecies due to differences in chromosomal organization and virulence. The subspecies *holarctica* (type B) is generally considered more clonal than the other subpopulations with moderate virulence compared to the hypervirulent A.I clade. We performed whole genome sequencing (WGS) on six type B strains isolated from the blood of patients with tularemia within a one-year period from the same United States region, to better understand the associated pathogenicity. The WGS data were compared to the prototype strain for this subspecies, specifically FSC200, which was isolated from a patient with tularemia in Europe. These findings revealed 520–528 single nucleotide polymorphisms (SNPs) between the six United States type B strains compared to FSC200, with slightly higher A+T content in the latter strain. In contrast, comparisons between the six type B isolates showed that five of the six type B isolates had only 4–22 SNPs, while one of the strains had 47–53 SNPs. Analysis of SNPs in the core genome for the six United States type B isolates and the FSC200 strain gave similar results, suggesting that some of these mutations may have been nonsynonymous, resulting in altered protein function and pathogenicity.

## 1. Introduction

The zoonotic disease tularemia is caused by the facultative intracellular pathogen *Francisella tularensis*. The high infectivity, the ability to cause a lethal disease, and the previous use as a bioweapon has resulted in the classification of this pathogen as a Tier 1 select agent by the Centers for Disease Control and Prevention (CDC) [1,2]. This Gram-negative coccobacillus can infect numerous invertebrate and vertebrate species, including humans [3]. Transmission of tularemia can occur from oral, cutaneous, and conjunctival routes of exposure to *F. tularensis*, as well as inhalation. The latter case causes pneumonia tularemia and is the most lethal form, with as few as 10 organisms causing mortality if untreated [4]. In the environment, common vectors for this pathogen include arthropods (i.e., ticks, deer flies, and mosquitoes), rodents, and lagomorphs, along with predatory animals such as cats [3,5]. *F. tularensis* has also been shown to survive in water and mud for several months [3]. However, the persistence and adaption of *F. tularensis* to various ecological niches in nature and numerous hosts is poorly understood.

The clinically relevant *F. tularensis* subspecies include subsp. *tularensis* (also referred to as type A) and subsp. *holarctica* (also referred to as type B). The subsp. *tularensis* (type A) has been further subdivided into subtypes A.I and A.II due to differences in genomic organization and virulence. A.I strains are some of the most pathogenic bacteria known, while *F. tularensis* A.II and B strains generally cause a milder case of tularemia in humans [6]. Furthermore, these *F. tularensis* subpopulations have geographic associations. Type A strains are primarily restricted to North America, with the A.II clade being located predominately in the western region of the United States. In comparison, type B isolates are prevalent throughout the Northern Hemisphere and are endemic in Europe [7].

We report herein the isolation and characterization of six *F. tularensis* type B strains from clinical cases of tularemia within a one-year period between 2014 and 2015. Importantly, all six of these type B isolates were cultured from the blood of different patients with tularemia and all six of these individuals resided in the same region within the United States, specifically Scottsbluff, Nebraska in the Midwest. To determine if these six *F. tularensis* type B isolates were clonal and possibly acquired additional virulence factors, whole genome sequencing (WGS) was performed and their content was compared to each other and to the most studied type B strain FSC200. Despite the same foci for these human tularemia cases, our findings revealed an unexpected number of single nucleotide polymorphisms (SNPs), indicating that the etiologic agent in these six blood infections was due to different type B strains. Despite the supposition that *F. tularensis* type B strains cause a milder form of tularemia, these finding indicate that this clade is capable of causing a serious infection that could result in sepsis and possibly death if untreated.

## 2. Materials and Methods

### 2.1. Bacterial Strains and DNA Isolation

The six *F. tularensis* subsp. *holarctica* (type B) strains were transferred to the University of Nebraska Medical Center (UNMC)/Nebraska Public Health Laboratory (NPHL) in Omaha following the requirements of the Federal Select Agent Program, as outlined in the Animal and Plant Health Inspection Service/CDC Form 2, Guidance Document for Request to Transfer Select Agents and Toxins [2]. Manipulation of viable culture material was performed by authorized individuals within a biosafety level 3 laboratory certified for select agent work by the United States Department of Health and Human Services, using laboratory biosafety criteria according to the requirements of the Federal Select Agent Program. The identity of these six wild-type isolates as *F. tularensis* was confirmed by qualified personnel at NPHL, a reference laboratory in the National Public Health Laboratory Network, using CDC-approved protocols.

All *Francisella* isolates were grown on chocolate agar plates (Remel, Lenexa, Kansas, KS, USA) and incubated at 37 °C with 5% CO_2_ for three days before processing for DNA extraction. Genomic DNA from the six *F. tularensis* strains was isolated using cetyltrimethyl ammonium bromide (CTAB), according to standard procedures [8]. High molecular weight genomic DNA was confirmed by fractionation in an agarose gel and visualization by subsequent ethidium bromide staining. DNA concentration was determined by measuring the absorbance at 260 nm using a Qubit 3.0 fluorometer (Invitrogen, Carlsbad, CA, USA). The purity of the genomic DNA was evaluated by obtaining the ratio of the absorbance at 260 nm and 280 nm (A_260/280_) utilizing a NanoDrop 1000 UV–visible spectrophotometer (Thermo Fisher, Waltham, MA, USA).

### 2.2. Whole Genome Sequencing (WGS)

*F. tularensis* genomic DNA libraries for the six different isolates were constructed using the Nextera XT Library Prep Kit (Illumina, San Diego, CA, USA), according to the manufacturer’s recommendations. WGS using 250 bp paired-end chemistry was performed on the Illumina MiSeq platform (Illumina) at NPHL, according to the manufacturer’s instructions. A rate of clusters passing the filter of > 80%, a Phred quality score (QS30) of > 75%, and a cluster density of 600 to 1300 were used as parameters to assess the quality of the run. FastQC 0.11 was used to evaluate the quality of the sequencing reads, and the reads were trimmed with Trimmomatic 0.33 [9]. FastQC 0.11 was then utilized to recheck read quality. Next, de novo assembly was performed using SPAdes 3.12 [10], and removal of contigs of less than 200 bp was carried out with BBMap 38.06 at http://sourceforge.net/projects/bbmap/. To assess the quality of the assembled genome, Quality Assessment Tool for Genome Assemblies QUAST 4.1 was utilized along with Python 2.7 [11].

### 2.3. Annotation of Genomes

The assembled genomes were annotated using Prokka 1.13 [12], and the National Center for Biotechnology Information (NCBI) Prokaryotic Genome Annotation Pipeline (PGAP) 4.11 (https://www.ncbi.nlm.nih.gov/genome/annotation_prok/) for each strain [13]. The genomes were evaluated for the presence of plasmids using PlasmidFinder 2.0 [14]. The assessment of acquired antibiotic resistance genes and virulence factors was performed using ResFinder 3.1 and VirulenceFinder 2.9 [15], respectively, which were available in the Center for Genomic Epidemiology server at http://www.genomicepidemiology.org/.

### 2.4. Comparison of Genomic Content

The CSI Phylogeny software was used to detect, filter, and then generate a SNP pairwise matrix for all the nucleotide differences present in the *F. tularensis* type B genomes. To generate phylogenetic trees, the files were converted to a Newick format, and rooted trees were created using the CSI Phylogeny tool. To visualize and annotate the dendrograms, the Interactive Tree of Life (iTOL) online tool at https://itol.embl.de/upload.cgi was utilized. Roary 3.12.0 was utilized to assess the pan-genome for the detection of the presence and absence of genes in the six sequenced wild-type isolates relative to the reference *F. tularensis* type B strain FSC200 (NCBI Reference Sequence NC_019551.1, GenBank accession number CP003862) [16]. Genome alignments were performed using the progressiveMauve software tool [17]. Default parameters were used for all software unless otherwise specified, and bioinformatic tools were utilized on the GitHub platform https://github.com/.

### 2.5. Data Availability

The whole genome sequences for the *F. tularensis* type B strains NE-RWMC-071715A, NE-RWMC-071715B, NE-RWMC-082115, NE-RWMC-083115, NE-RWMC-101314, and NE-RWMC-101714 were deposited in NCBI BioProject number PRJNA663584 and are associated with BioSample numbers SAMN16136936, SAMN16136935, SAMN16136934, SAMN16136933, SAMN16136932, and SAMN16136931; Sequence Read Archive (SRA) numbers SRR12644162, SRR12644163, SRR12644164, SRR12644165, SRR12644166, and SRR12644167; and GenBank accession numbers JACWLF000000000, JACWLG000000000, JACWLH000000000, JACWLI000000000, JACWLJ000000000, and JACWLK000000000, respectively. The strain designations for these six wild-type *F. tularensis* isolates denote the abbreviation for the associated state (i.e., Nebraska) and hospital (i.e., Regional Western Medical Center), and the six digit numbers (i.e., mmddyy) denote the month, day, and year that the wild-type strain was archived into the UNMC/NPHL database and stock collection, which was around three days after the initial isolation of the associated pathogen.

## 3. Results

Six patients admitted to the same hospital in western Nebraska (longitude W-103.662024, latitude N 41.867089) between October 2014 and August 2015 had symptoms of an infection, which included fever, chills, nausea, headache, hypotension, and an elevated white blood cell count. These individuals were given antibiotics to prevent further complications, based on their clinical presentation. Blood drawn from these patients prior to antibiotic treatment was cultured on various media, using different growth conditions and incubation lengths (1–3 days). Next, the criteria established by the CDC national Laboratory Response Network for the initial identification of a potential *F. tularensis* isolate were implemented. This process involves a rule-in/rule-out process that includes the visualization of colony morphology, Gram staining, and biochemical testing. All six wild-type isolates required a minimum of 2 days for visible growth on medium that contained a sulfhydryl source (i.e., cysteine or cystine). Morphological examinations of these microbes revealed a nonmotile, nonsporulating, Gram-negative coccobacillus, and biochemical testing showed that all six of these isolates were oxidase negative.

To confirm the identification of the infectious blood-derived pathogens as *F. tularensis*, the cultures were transported to NPHL. Real-time PCR assays recommended by the CDC were performed and all three *F. tularensis* gene targets (FT1, FT2, and FT3) were detected and amplified in each of the six wild-type isolates. These assays provided qualitative detection of *F. tularensis* genomic DNA, but only presumptive identification. To confirm the identification of these isolates, direct fluorescence antibody (DFA) analyses from culture were performed using a rabbit polyclonal antibody that detects an *F. tularensis*-specific antigen. All six wild-type isolates were positive in the DFA test. Additional PCR-based assays were then performed for typing of the *F. tularensis* isolates, as previously described [18,19]. These analyses determined that all six of these blood-derived pathogens were *F. tularensis* subspecies *holarctica* (type B) in these clinical cases. A stock culture of these six select agent strains was then made by authorized individuals for long-term storage in the Select Agent-registered UNMC/NPHL high-containment BSL-3 facility.

To further evaluate these six wild-type *F. tularensis* isolates, the genomic DNA from these pathogens was extracted and sequenced. The assembly of the six *F. tularensis* genomes resulted in 98 to 101 contiguous sequences or contigs, with the longest fragment containing 88,291 bp. The total length obtained for the six bacterial chromosomes ranged from 1,787,790 to 1,790,384 bp, which translates to an estimated coverage of 94.5%. N50 lengths are often used as a measure of assembly quality, capturing how much of the genome is covered by relatively large contigs [20]. The N50 (bp) acquired for the contigs associated with the six *F. tularensis* genomes was between 27001–25313, which statistically conveys high assembly quality in terms of contiguity. The average G+C content for these six genomes was determined to be 32.16–32.18%, which is similar to the majority of other select agent strains in this species [21,22].

Next, the genomic content for these six *F. tularensis* type B isolates was compared to each other and to the reference type B strain FSC200. *F. tularensis* FSC200 was isolated from a child in 1998 that lived in an area in Sweden where tularemia is endemic due to infections by this subspecies and is the most studied type B select agent strain [4,23]. Sequence comparison of the six *F. tularensis* type B strains to FSC200 indicated that no additional plasmids, antibiotic resistance genes, or virulence factors were acquired. The chromosomal alignments shown in Figure 1 compare the local collinear blocks (LCBs) that were obtained from the WGS of the six type B strains relative to each other and to FSC200. These results showed that the six *F. tularensis* type B strains shared similar nucleotide content to each other (Figure 1A), as well as to FSC200 (Figure 1B,C). However, in order to compare overall genomic content and organization, complete chromosomal assembly is required, but the numerous insertion sequence elements and repeated sequences within the *F. tularensis* select agent strains make this an arduous and time-consuming task.

Table 1 compares the SNP content in the WGS obtained for the six *F. tularensis* type B strains that were isolated from the blood of patients in the United States during a one-year period to each other and to the prototype strain FSC200. The six United States strains showed substantially higher homology to each other than to the type B reference FSC200, with more than 519 SNP differences identified between these geographically separated isolates. Five of the six *F. tularensis* United States type B isolates had fewer than 23 SNPs, indicating that these strains were closely related to each other. In contrast, RWMC-082115 had more than 46 SNPs compared to the other five United States isolates (Table 1). RWMC-101314 and RWMC-071715A shared the highest WGS identity with only four SNPs, even though these pathogens were isolated from the blood of distinct patients in two different years. However, these four SNPs were located at different regions within the chromosome of these two strains, and only RWMC-071715A shared these polymorphisms with FSC200. Although RWMC-071715A and RWMC-071715B were isolated on the same day from two separate individuals at the same hospital, WGS results showed a difference of 22 nucleotides between these two strains (Table 1). Together, these results indicated that despite these nucleotide differences, these six *F. tularensis* type B strains maintained the ability to cause the highly infectious disease tularemia.

In order to compare the genomic content of genes shared between the six *F. tularensis* type B strains isolated from the same region in the United States and reference strain FSC200, pairwise matrix SNP content in the core genome was obtained and evaluated (Table 2). These data showed similar results to what was obtained for the WGS pairwise matrix SNP comparisons, which is shown in Table 1. However, there were slightly fewer SNP differences in shared genes in the six United States type B and FSC200 genomes, compared to the SNP content obtained for the WGS comparisons. These results suggested that these nucleotide differences are within coding regions, which could possibly alter the function of the gene products if these mutations are nonsynonymous. Furthermore, there was again a considerably higher number of core genome SNPs (> 518) in the six United States type B strains relative to FSC200 (Table 2), which was similar to the 519 SNPs obtained for the WGS comparisons (Table 1). RWMC-101314 and RWMC-071715A, which shared the highest WGS identity with only four SNPs, again only had four SNPs when comparing the core genome for these two strains, further indicating clonal relatedness between these 2014 and 2015 isolates, respectively.

The clustering patterns shown in Figure 2 for the rooted tree were obtained for both the WGS and core genome SNP comparison, and further depict the close relatedness of the six United States type B isolates, and the considerable divergence of these strains from *F. tularensis* FSC200. Although these six United States strains shared considerable genomic content to each other, the higher number of SNPs in RWMC-082115 compared to the five other strains from this region resulted in a separate branch in the phylogenetic tree (Figure 2). However, no obvious differences were observed in the clinical presentation caused by RWMC-082115 compared to these other United States type B strains. Nucleotide composition comparisons of the SNPs in the aforementioned *F. tularensis* strains revealed that FSC200 had a slightly higher A+T content (3%) compared to the six United States type B strains.

## 4. Discussion

The positive identification of *F. tularensis* type B in peripheral blood obtained from patients with symptoms of an infection (i.e., fever, chills, and nausea) was confirmed in six different clinical cases from the same region within a one-year period. These bloodstream infections could have led to septic shock, organ failure, and then ultimately death had immediate antibiotic treatment not been given to these patients. Although the majority of tularemia cases in the United States are due to the hypervirulent subtype A.I clade and not type B strains [6], the current report emphasizes that all select agent strains from this species are capable of causing bacteremia and potentially mortality. Furthermore, the comparison of these isolates with the prototype *F. tularensis* FSC200 strain from Europe revealed a substantial number of SNPs. These findings suggest that although the type B clade is considered more clonal than the A.I and A.II strains [18,24], this subpopulation is still highly pathogenic despite the apparent accumulation of SNPs.

In 2015, the CDC reported on a substantial increase in the human cases of tularemia in Nebraska, Colorado, Wyoming, and South Dakota [25]. These clinical cases included the four type B strains described in this communication, specifically RWMC-071715A, RWMC-071715B, RWMC-082115, and RWMC-083115. In this tularemia outbreak, some of the infected individuals suggested two or more possible causes; however, the source of *F. tularensis* was never verified. Although the source of *F. tularensis* in clinical cases is often unknown, hard ticks are generally considered the main biological vector of tularemia in North America [26,27]. However, other infected organisms, especially predatory animals [5], along with contaminated air, food, and water, can also be responsible for transmitting this pathogen to another vector or host [3]. Therefore, the multidirectional nature of zoonosis and the broad host range of *F. tularensis* have complicated epidemiological investigations for the identification of tularemia foci in the environment.

Although the *F. tularensis* subspecies *holarctica* (type B) strains have fewer genomic rearrangements than the other clinically relevant subspecies of *tularensis* [18,24], erythromycin sensitivity versus resistance biovars, canonical SNP (canSNP) analyses, and chromosomal content have been used to identify and classify type B strains into various subgroups [18,19,28,29]. Karlsson and associates determined that the A2059C SNP in the 23S ribosomal RNA gene was conserved in biovar II strains and conferred the associated resistance to erythromycin [28]. Since this SNP is also found in the most abundant subgroup B.12 in Eurasia that includes FSC200, the authors suggested that this clonal inheritance may have contributed to the success of this phylogenetic group. The six type B strains analyzed in this report did not have the A2059C SNP in the 23S ribosomal RNA gene, but rather had two other mutations (G449A and C2052A) compared to FSC200, suggesting that other factors have influenced the prevalence of these pathogenic isolates.

Many bacteria subjected to relaxed selection are often intracellular and, in general, reside or replicate within the infected host cell with a limited chance of recombining or exchanging DNA with other bacteria [30,31,32]. Moreover, GC to AT mutations have been shown to occur approximately twice as often as AT to GC mutations and seem to be more readily fixed within the genome [33,34]. The slightly higher A+T content in *F. tularensis* FSC200 compared to the six United States type B strains described in this report suggests that this European isolate was subjected to more conditions that provided relaxed purification, which led to the accumulation of these AT mutations. Conversely, one cannot rule out the possibility that the United States type B strains may have encountered stringent selection, which prevented or reduced G/C to A/T allelic modifications.

The analysis of canSNPs and canonical insertions/deletions (canINDELs) in *F. tularensis* genomes has certainly improved the ability to correlate lineages and geographical associations [35,36,37,38]. However, Johannsson and associates determined that a tularemia outbreak in Europe was caused by diverse type B strains with no significant correlation between genetic, spatial, or temporal distances [39]. Therefore, additional study is needed to provide a better understanding of the mechanisms that allow *F. tularensis* to persist in nature and evade immune surveillance during an infection.

## Figures and Tables

**Figure 1 microorganisms-08-01515-f001:**
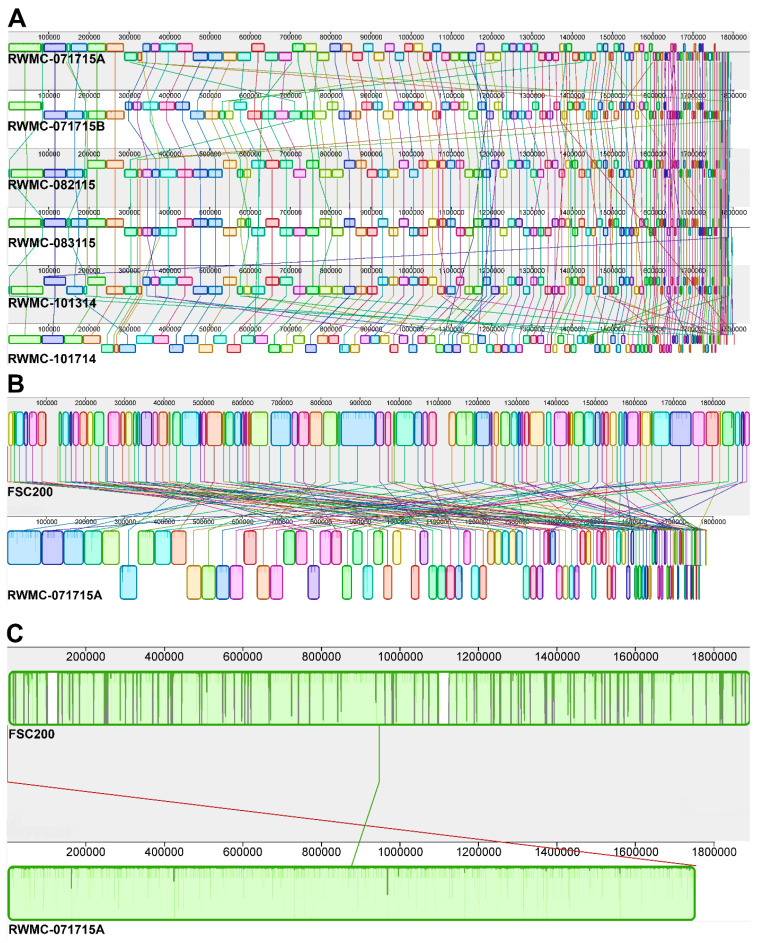
Diagram depicting local collinear blocks (LCBs) within the genomes of the six wild-type *F. tularensis* type B isolates compared to the reference type B strain FSC200. (**A**) Comparison of the LCBs obtained for *F. tularensis* strains RWMC-071715A, RWMC-071715B, RWMC-082115, RWMC-083115, RWMC-101314, and RWMC-101714. (**B**), (**C**) Comparison of the LCBs for a representative *F. tularensis* type B strain (RWMC-071715A) with the reference type B strain FSC200 (NCBI reference sequence NC_019551.1, GenBank accession number CP003862), before and after contig correlation to FSC200, respectively. The large white LCBs in FSC200 denote missing WGS data in RWMC-071715A in Figure 1C. These regions at nucleotide positions 100,370–129,503 and 1,096,979–1,126,076 in FSC200 contained two additional copies of the genes encoding 16S ribosomal RNA (rRNA), 23S rRNA, 5S rRNA, and transfer RNA, as well as insertion sequence elements and hypothetical genes. Contigs less than 128 bp were not included in the comparison to FSC200 in Figure 1B,C. Each LCB is represented with a different pattern and/or shading. An LCB depicted below the center line indicates that the sequence is reverse/complement, and the relative nucleotide lengths are denoted in base pairs above the corresponding genome. The progressiveMauve software tool was used to align the genomes [17].

**Figure 2 microorganisms-08-01515-f002:**
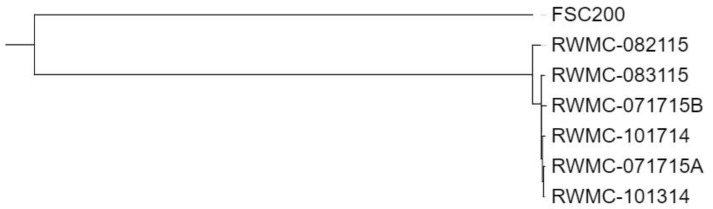
Phylogenetic analysis of the WGS and core genome SNPs obtained for the six wild-type *F*. *tularensis* type B strains isolated from the blood of infected individuals from the same region in the United States with tularemia, along with prototype strain FSC200 for comparison. The rooted dendrogram was generated with the CSI Phylogeny tool and the resulting SNP-based tree was obtained for both analyses. A correlation coefficient of 0.99 was obtained for this analysis.

**Table 1 microorganisms-08-01515-t001:** Whole genome sequencing (WGS) single nucleotide polymorphism (SNP) pairwise matrix for the six *F. tularensis* type B strains isolated from the same region in the United States within a one-year period relative to each other and to the prototype type B strain FSC200 ^1,2^.

	FSC200	071715A	071715B	082115	083115	101314	101714
FSC200	0	528	525	524	526	526	520
071715A	528	0	22	49	17	4	12
071715B	525	22	0	53	21	20	22
082115	524	49	53	0	48	47	49
083115	526	17	21	48	0	15	17
101314	526	4	20	47	15	0	10
101714	520	12	22	49	17	10	0

^1^ The six *F. tularensis* type B strains from the United States are denoted without the prefix RWMC. ^2^ The SNP pairwise matrix was generated using the CSI Phylogeny tool.

**Table 2 microorganisms-08-01515-t002:** Single nucleotide polymorphism (SNP) pairwise matrix for the core genome from the six *F. tularensis* type B strains isolated from the same region in the United States within a one-year period relative to each other and to the prototype type B strain FSC200 ^1,2^.

	FSC200	071715A	071715B	082115	083115	101314	101714
FSC200	0	523	524	520	522	521	519
071715A	523	0	17	47	15	4	4
071715B	524	17	0	48	16	15	13
082115	520	47	48	0	46	45	43
083115	522	15	16	46	0	13	11
101314	521	4	15	45	13	0	2
101714	519	4	13	43	11	2	0

^1^ The six *F. tularensis* type B strains from the United States are denoted without the prefix RWMC. ^2^ The SNP pairwise matrix was generated using the CSI Phylogeny tool.

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
