# Peer review of "Differences in Blood-Derived Francisella tularensis Type B Strains from Clinical Cases of Tularemia"

_microorganisms, 2020, doi:10.3390/microorganisms8101515_

Round 1

Reviewer 1 Report

Larson et al. performed whole genome sequencing on six clinical isolates of F. tularensis Type B strains and compared their sequences to that of a Type B reference strain. The manuscript is well written; however, a few summary statements do not seem to be supported by the data presented especially when a single Type B reference genome was used as a basis for the authors' main conclusions and only six isolates in the United States were analyzed and used to support their evolution and divergence from European strains, which is only represented by one reference strain.

Lines 179-180 the six sequenced strains "are slowly evolving to their environmental niche"; lines 211-212 "these findings indicated that the...strains have evolved numerous SNP differences in the United States compared to this clade in Europe"; and lines 230-231 "this subpopulation is slowly evolving without any apparent loss in pathogenicity": the environmental sources of the isolates were not identified and the evolution conclusion is based on sequence comparisons to one reference genome and analysis of six clinical isolates from Nebraska.

line 112: missing sequence database numbers
lines 115-117: it would be useful for the authors to define the strain designations, e.g. the 6 digit strain number denotes mmddyy of isolation
lines 137-138: G+C content statement needs a reference
Figure 1: text in the images need to be higher resolution so they can be read
lines 173-174: check phrasing for "...differed the most relative..."
Table 1 and lines 178-179: authors stated 071715A and B had the highest # of SNPs compared to the other strains except 082115, but what about 101314
lines 210-215: These lines should be in the discussion with references and an expansion of the "clade" comment in line 212.
Figure 2: why is 082115 on a separate branch compared to the other 5 isolates other than the higher number of SNPs? e.g. was there any clinical aspects that were unique to this strain absent in the others that are not specifically described in lines 115-117?
line 257: authors indicated the possibility of G+C going to A+T to explain the higher A+T content in the reference genome ("relaxed selection"), but why not also A+T going to G+C in the six Nebraska isolates, perhaps stringent selection?

Reviewer 2 Report

The authors of the manuscript " Differences in Blood-derived Francisella tularensis  Type B Strains from Clinical Cases of Tularemia " describe the isolation of six clinical F. tularensis. subsp. holarctica strains and results of whole genome sequencing (WGS) performed on these strains

Despite all the advantages of this article, I would like to make a few small comments.

General Comment: At the time of writing the review, authors did not provide open access to the WGS data. Therefore, it is not possible to assess the technical aspects of the work.

Lines 121-129

Maybe, it would be worthwhile to describe in more detail the morphological and biochemical properties of the isolates, and also indicate which PCR tests were performed (target genes with links to the literature source, or the names of PCR test systems)

Lines 233-241

It isn’t necessary to discuss there the epidemiological role of ticks. Tularemia can be transmitted in many ways besides tick bites. In the clinical cases described here, the source of infection is designated as unknown. Thus, we can conclude that there are no reasons to believe that authors describe tick-borne tularemia, or, more generally arthropod-borne tularemia. This paragraph should either be supported by a clinical history indicating the source of the disease (e.g. skin lesions associated with tick-borne tularemia, or patient testimony that they were bitten by a tick at a certain period before infection, etc.) or deleted. Or it can be expanded a little and moved to the "introduction" section

Lines 242-243

It is better to slightly change the phrase "Although the F. tularensis subspecies holarctica (type B) strains have fewer genomic rearrangements than the other subspecies" In the works that the authors cite to confirm this thesis ([20,21]), mainly the genetic diversity of two main subspecies holarctica and tularensis is discussed. There is too little data on the other two subspecies, mediaasiatica and novicida to reasonably discuss their genetic polymorphism. So we suggest changing "... other subspecies" to ".... subsp. tularensis "

Lines 259-260

The phrase "The wider geographic dispersion of F. tularensis type B strains compared to the other F. tularensis select agent clades supports the notion that this subpopulation has a higher capacity to survive in different environmental niches." is too speculative and too categorical. We are not aware of any data that would indicate that the subsp. holarctica is more adaptive and/or capable of infecting a wider range of hosts than the subsp. tularensis. Authors should add appropriate links, or remove this phrase.

Round 2

Reviewer 1 Report

I thank the authors for addressing my comments and concerns.